# Is Forward Gradient an Effective Tool for Explaining Black-box Models?

## Abstract

Gradients are widely used to explain the decisions of deep neural networks. However, as models become deeper and more complex, computing gradients become challenging and sometimes infeasible, hindering traditional explanation methods. Recently, the forward gradient method has garnered attention for training structure-agnostic models with discontinuous objective functions. This method perturbs only the parameters of interest for gradient computation and optimization. Inspired by this, we investigate whether the forward gradient can be employed to explain black-box models. In this work, we use the likelihood ratio method to estimate output-to-input gradients and utilize them for the explanation of model decisions. Additionally, we propose block-wise computation techniques to enhance estimation accuracy. Extensive experiments to explain various models, including CNNs, LSTMs, Phi-3, and CLIP, in black-box settings validate the effectiveness of our method, demonstrating good gradient estimation and improved explainability under the black-box setting.

## 1 Introduction

Deep Neural Networks (DNNs) have achieved remarkable success across a range of applications, including autonomous driving (Grigorescu et al., 2020; Mozaffari et al., 2020; Huang & Chen, 2020), facial recognition (Mehdipour Ghazi & Kemal Ekenel, 2016; Fathallah et al., 2017; Mellouk & Handouzi, 2020), and clinical diagnostics (De Fauw et al., 2018; Van der Laak et al., 2021; Kermany et al., 2018). In these safety-critical domains, it is imperative that DNNs not only deliver high task performance but also provide transparent and understandable decision-making processes (Tambon et al., 2022; Borg et al., 2018). Extensive research has been devoted to demystifying DNNs, exploring various approaches such as counterfactual explanations (Mothilal et al., 2020; Slack et al., 2021), attribution and activation analysis (Achtibat et al., 2023; Lin et al., 2021), and saliency maps (Hu et al., 2023; Tjoa & Cuntai, 2022; Lorentz et al., 2021). Of these, gradients play a crucial role in decision explanation because of their robust theoretical foundations (Khorram et al., 2021;

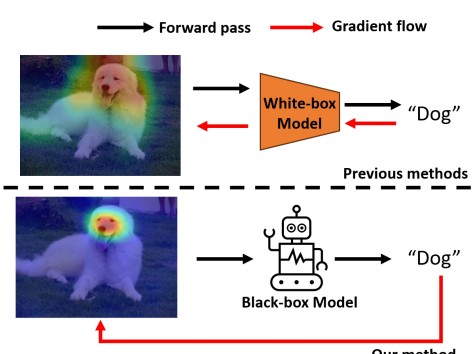

Figure 1: Previous methods require the full knowledge of studied models to compute the gradient for model explanation, which makes it impossible to be applied to explain the black-box model. Our proposed method only needs one forward pass of the studied model. It can directly get the gradient without relying on the backward pass and the knowledge of model architectures, thus enabling the black-box model decision.

Kapishnikov et al., 2021) and superior performance. Moreover, gradient-based explanation methods are data-centric, offering greater adaptability to different model architectures. This flexibility is particularly promising in elucidating the decisions from black-box models, which are either proprietary and cloud-based (OpenAI, 2020) or computationally intensive (Radford et al., 2021).

The use of gradients in model explanation can be broadly divided into two categories: class activation mapping (Zhou et al., 2016; Chattopadhay et al., 2018; Wang et al., 2020; Omeiza et al., 2019) and saliency map (Sundararajan et al., 2017; Erion et al., 2021; Lundstrom et al., 2022; Yang et al., 2023). Methods based on class activation map up-sample the feature map inner models for model explanation. Gradient-based methods derive the gradients of inputs relative to the output, thereby highlighting the model's areas of interest. These two approaches are intuitive and efficient but become infeasible when the model architecture is inaccessible.

In black-box machine learning scenarios (Bodria et al., 2023), where access to full knowledge of computation graphs for optimization is unavailable, the problem becomes a Zeroth-order Optimization (ZO) problem (Chen et al., 2017). Among various ZO strategies (Chen et al., 2019; Rando et al., 2024; Kozak et al., 2023; Vemula et al., 2019), the likelihood ratio method (Peng et al., 2022) stands out. It pushes the parameters out of the loss function by introducing noise, allowing for unbiased gradient estimation. This method has shown competitive performance in training diverse neural networks, comparable to that achieved with backpropagation (Rumelhart et al., 1986). Such promising results inspire us to think about the plausibility of the likelihood ratio method for gradient estimation on inputs, facilitating black-box explanations through saliency maps. Motivated by this, we propose to use the likelihood ratio method to bridge the gap between gradient-based explanation methods and black-box scenarios, as illustrated in Fig. fig. 1, where we skip the access of backward pass of the model to get the gradient and explain the model decision.

Applying the likelihood ratio method to develop interpretation methods under black-box conditions is a non-trivial task, presenting two significant challenges. First, there is the question of how to estimate gradients on various characteristics of data using the likelihood ratio method. Many existing studies on zeroth-order optimization assume full knowledge of the local computational structure (Peng et al., 2022; Jiang et al., 2023), a luxury not available in our context where the whole model's structure, even the first layer for processing the input, is unknown. Second, we must address how to minimize the estimation variance of gradients. The likelihood ratio method, by its nature of noise injection, tends to have a high variance in gradient estimation (Peng et al., 2022; Jiang et al., 2023), a problem exacerbated by the high dimensionality, such as those in the $\mathbb{R}^{224 \times 224 \times 3}$ space of the ImageNet-1K dataset. Accurately estimating gradients in such high-dimensional spaces is challenging.

In this study, we introduce a unified framework for explaining black models using gradients. Our approach begins with deriving estimated gradients of inputs under a black-box model set using the likelihood ratio method. Then, we propose a blockwise computation pattern to address the challenge of high variance in gradient estimation. In evaluation, we integrate our framework to explain various models, including a series of computer vision models, text classification models, and the large language model Phi (Abdin et al., 2024). We also verify the feasibility and generalization ability in explaining multi-modal models against spurious correlations. There are three key findings revealed in our work which can be summarized as follows:

- ✅**Interpretability of forward gradients**: Compared to vanilla gradients, forward gradients offer comparable interpretability with greater scalability in explaining model decisions.
- ✅**Bridging the explanation gap**: Forward gradients provide a crucial link between white-box explanation methods and the challenges of explaining black-box models.
- ✖**High computation and memory-consumption**: Despite their explanatory power, forward gradients suffer from high computational and memory costs, due to large estimation variance. This issue can only be mitigated by using a large number of copies during computation.

## 2 RELATED WORK

### 2.1 GRADIENTS FOR MODEL EXPLANATION

Gradients are widely used in explaining models, including the class activation mapping (CAM) (Zhou et al., 2016; Chattopadhay et al., 2018; Wang et al., 2020; Omeiza et al., 2019) and the saliency map Sundararajan et al. (2017); Erion et al. (2021); Lundstrom et al. (2022); Yang et al. (2023).

CAM is first introduced in (Zhou et al., 2016), which employs the global average pooling neuron activations to weakly localize the objects in the inputs for saliency mapping. The GradCAM (Selvaraju et al., 2017) inherits the idea of CAM and combines the model gradients with the activations of

its internal neurons to compute the saliency maps. The GradCAM has been adopted and improved by a lot of later works. Such as GradCAM++(Chattopadhay et al., 2018), SmoothGrad(Smilkov et al., 2017), Smooth-Grad++ (Omeiza et al., 2019), and Grad-CAM++(Chattopadhay et al., 2018) to further boost the visualization ability. CAM requires the full knowledge of model architecture and feature maps during the inference process, thus failing to work under the black-box setting(Belharbi et al., 2022; Linardatos et al., 2020; Jiang et al., 2021).

The generation of saliency maps (Sundararajan et al., 2017; 2016) requires the gradients of the decision as inputs, which treat the model as a unified module regardless of the inner structure. Compared with directly leveraging the vanilla gradients, the use of integrate gradients is more popular. These methods gather the gradients of images along a path from the input to a reference image. The reference image represents what it looks like when the features present in the input are missing. Sundararajan et al. (2017) first proposed the use of the path attribution strategy, which can preserve certain desirable axiomatic properties for the explanations. This work inspires others to further investigations (Erion et al., 2021; Lundstrom et al., 2022; Yang et al., 2023).

### 2.2 ZEROTH-ORDER OPTIMIZATION

Zeroth-order optimization (Wang et al., 2018; Golovin et al., 2019; Chen et al., 2019) is proposed to address challenges in optimizing the non-differential criteria. It typically uses random perturbation on input to estimate the gradient and follows the stochastic gradient descent to update the parameters of interest. Popular zeroth-order optimization methods include the simultaneous perturbation stochastic approximation (SPSA) (Spall, 2000; 1997; Maryak & Chin, 2001; Granichin & Amelina, 2014), evolutional strategy (ES) (Salimans et al., 2017), and likelihood ratio (LR) (Peng et al., 2022; Jiang et al., 2023). SPSA individually perturbs each parameter and estimates the gradients. The low efficiency largely hinders its application in large-scale optimization problems. ES and LR are proposed to optimize deep learning models, which inject the noise into neuron weights and outputs respectively. Among these, LR stands out for its compelling performance in accurate gradient estimation (Jiang et al., 2023). In our work, we employ the LR to compute the gradient without reliance on the chain-rule to explain model decisions.

## 3 METHODOLOGY

We take the crafting of saliency map in computer vision as an example of using gradients to explain the model decision. In section 3.1, we present the preliminaries of the saliency map and why the application to black-box models is limited. In section 3.2, we introduce our framework based on likelihood ratio gradient estimation to explain black-box models. In section 3.3, we propose the blockwise computation manner that effectively enhances the quality of the saliency map generation, especially for high-dimensional inputs and large black-box models.

### 3.1 PRELIMINARIES

**Functions of gradients as saliency maps**. Given any image-label pair example $(x, y) \in \mathcal{X} \times \mathcal{Y}$, with $\mathcal{X} \subset \mathbb{R}^d$ representing the domain of input images and $\mathcal{Y} = \{1, ..., C\}$ denoting the set of possible label classes, let us consider a classification neural network, $f \colon \mathcal{X} \to \mathbb{R}^C$, which assigns a predicted class activation $\hat{y}_c = f_c(x)$ for each class $c \in \mathcal{Y}$. Numerous studies Sundararajan et al. (2017); Erion et al. (2021); Lundstrom et al. (2022); Yang et al. (2023) have investigated the use of gradient-based saliency maps to visualize and elucidate the predictive mechanisms of the network $f$. Provided that the neural network $f$ is differentiable almost everywhere, a gradient-based saliency map for a class $c$ can be represented as a function dependent on the gradients of $f$ w.r.t. inputs at dimension $c$, and a particular input instance $x_0$ as follows,

$$M_c(f, x_0) = S(g_c, x_0), \tag{1}$$

where $g_c(x_0) = \nabla_x f_c(x)|_{x=x_0}$ is the gradients of $f$ w.r.t. inputs at $x_0$.

**Inaccessibility of black-box models' gradients**. The gradients of neural networks w.r.t. inputs are critical, as they quantify the impact of a tiny change in the input on the output and, therefore, can be linked intuitively to the importance of various input features. However, the reliance on these gradients poses a significant challenge in the context of black-box models, where the gradients are inaccessible

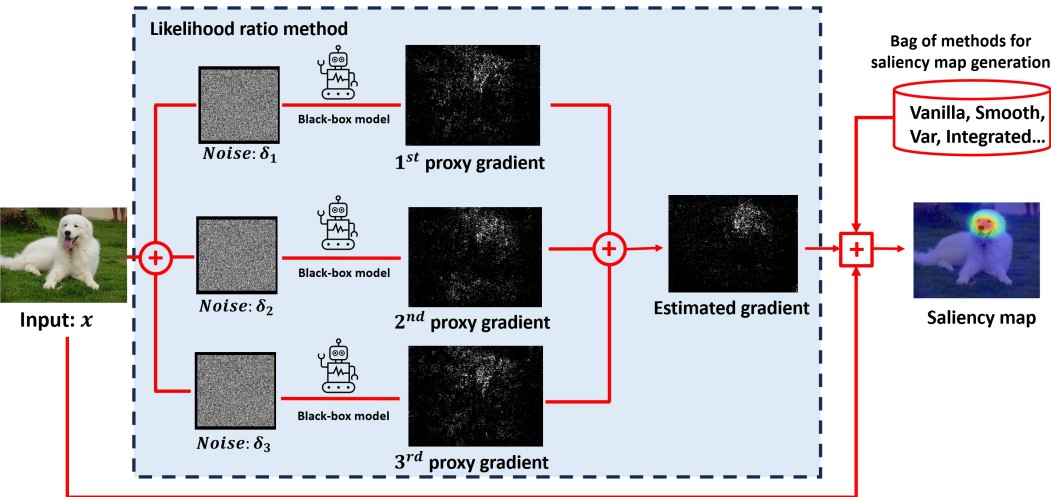

Figure 2: While the gradients of the black-box model can not be directly fetched, we use the likelihood ratio method to compute the proxy gradient of each noisy copy and obtain the estimated gradient. We incorporate the estimated gradient with gradient-based saliency map generation methods for the model explanation.

or even the internal workings are obscured. In fact, any use of black-box modules in the model would obstruct gradient computation due to the inapplicability of the chain rule, a fundamental principle in differential calculus. This limitation severely restricts the application of gradient-based saliency map techniques in the black-box setting. Therefore, the key problem in leveraging saliency map methods to interpret such models is to escape the inaccessibility of the black-box module to compute the gradients w.r.t. inputs.

## 3.2 LIKELIHOOD RATIO METHOD FOR SALIENCY MAP OF BLACK-BOX MODELS

In this section, we introduce a novel, unified framework for generating gradient-based saliency maps to interpret the decision of black-box models. As depicted in Fig. fig. 2, our framework encompasses three steps. Initially, we introduce perturbations to the images by injecting the noise before the model's forward pass. Subsequently, we compute the proxy gradient of each perturbed copy respectively and then average them to approximate the true gradient. Finally, we employ the estimated gradient to craft the saliency map for the model explanation. The following paragraphs will detail the three steps in our proposed framework.

**Injecting noise to inputs**. Within the context of a black-box model $f$, which is differentiable almost everywhere but whose internal gradients are inaccessible, consider that we employ a technique of noise injection, in which we add small random noise $z$ into the input $x$. Intuitively, if the noise has a neutral mean (zero) and a variance small enough, the expectation of gradient *w.r.t.* the noise-added input, $\mathbb{E}_z(g_c(x+z))$, would be close enough to the true gradient $g_c(x)$. Practically, one can easily verify that as the standard deviation $\sigma$ of the noise approaches zero, the expectation of gradient w.r.t. the noise-added input converges to the true gradient, *i.e.*, $\lim_{\sigma \to 0^+} \mathbb{E}_z(g(x+z)) = g(x)$, implying the use of $\mathbb{E}_z(g_c(x+z))$ as a viable approximation of $g_c(x)$. This technique allows us to control the precision loss of expected noise-adding gradients by selecting noise $z$ with a distribution close to 0. In many situations, such as the Gaussian noise $\mathcal{N}(0, \sigma^2\mathbb{I})$, it is equivalent to adjusting the $\sigma$.

**Likelihood ratio gradient estimator**. Directly computing $\mathbb{E}_z(g_c(x+z))$ still necessitates access to the gradient, which exactly contradicts our objective of addressing the black-box setting where such information is unavailable. Therefore, we propose our likelihood ratio gradient estimator for black-box models. Initially, by forwarding the noise-added input into the model, we obtain the corresponding class activation $f_c(x_0+z)$. Then, we compute the proxy gradient for each noise-added input by multiplying the class activation with the negative gradient of the noise's log probability density function. Finally, we average $n$ samples of proxy gradients to form our likelihood ratio

gradient estimator. Mathematically, this can be formalized as:

$$\hat{g}_c^{\text{LR}}(x_0) := \frac{1}{n} \sum_i^n (-f_c(x_0 + z_i) \nabla_z \ln \mu_z(z_i)) \tag{2}$$

where $n$ is the number of perturbed sample, *i.e.*, copies, $\{z_i\}_{i=1}^n$ represents the i.i.d. random noise, and $\mu_z(\cdot)$ denotes the probability density function of the injected $z$. Notably, when the injected noise $z_i \overset{\text{iid}}{\sim} \mathcal{N}(0, \sigma^2 \mathbb{I})$, the likelihood ratio gradient estimator can be simplified into $\hat{g}_c^{\text{LR}}(x_0) = \frac{1}{n\sigma^2} \sum_i^n f_c(x_0 + z_i) z_i$.

Additionally, we introduce the following theorem 1, establishing the foundation for our likelihood ratio gradient estimator for black-box models.

**Theorem 1** *Assume that $\lim_{|\zeta| \to \infty} f_c(x_0 + \zeta) \mu_z(\zeta) = 0$ for any input $x_0$. Let $z$ denote the random noise with the same distribution as $\{z_i\}_1^n$. Then, we have*

$$\mathbb{E}(\hat{g}_c^{LR}(x_0)) = \mathbb{E}_z(g_c(x_0 + z)). \tag{3}$$

**Proof 1** *Notice that a direct corollary from $\lim_{|\zeta| \to \infty} f_c(x_0 + \zeta) \mu_z(\zeta) = 0$ is*

$$\int_{\mathbb{R}^d} \nabla_\zeta f_c(x_0 + \zeta) \mu_z(\zeta) d\zeta + \int_{\mathbb{R}^d} f_c(x_0 + \zeta) \nabla_\zeta \mu_z(\zeta) d\zeta$$

$$= \int_{\mathbb{R}^d} \nabla_\zeta \left( f_c(x_0 + \zeta) \mu_z(\zeta) \right) d\zeta = 0. \tag{4}$$

*Therefore, we can derive*

$$\begin{aligned}
\mathbb{E}_z(g_c(x_0 + z)) &= \int_{\mathbb{R}^d} \nabla_x f_c(x)|_{x=x_0+\zeta} \mu_z(\zeta) d\zeta & \triangleright \textit{by definition} \\
&= \int_{\mathbb{R}^d} \nabla_\zeta f_c(x_0 + \zeta) \mu_z(\zeta) d\zeta & \triangleright \textit{by change of variable} \\
&= -\int_{\mathbb{R}^d} f(x_0 + \zeta) \nabla_\zeta \mu_z(\zeta) d\zeta & \triangleright \textit{by eq. (4)} \\
&= \mathbb{E}_z(-f_c(x_0 + z) \nabla_z \ln \mu_z(z)) & \triangleright \textit{by derivative of } \ln \\
&= \mathbb{E}(\hat{g}_c^{LR}(x_0)). & \triangleright \textit{by noise}
\end{aligned}$$

theorem 1 indicates that the expectation of our proposed gradient estimator is the same as the expectation of gradient w.r.t. the noise-added input under a certain asymptotic growth condition, which is commonly satisfied in practice. This, together with the previous discussion about $\mathbb{E}_z(g_c(x_0 + z))$, substantiates the feasibility of utilizing $\hat{g}_c^{\text{LR}}$ as an estimator of the true gradient. While this approach introduces a certain bias, as discussed, such impacts can be effectively mitigated by carefully controlling the standard deviation of the injected noise.

**Integrating with gradient-based saliency map methods**. By injecting random noise into the input alongside our likelihood ratio gradient estimator, we successfully circumvent the reliance on direct access to the gradients of black-box models. This approach allows for the seamless integration of any gradient-based saliency map technique within the context of black-box models.

To illustrate, consider an arbitrary gradient-based method that generates saliency map $M_c$ for a model $f$ and an image $x$ from a function $S$ of the model's gradient w.r.t. inputs and that image, $M_c(f, x) = S(g_c, x)$, as depicted in eq. (1). For instance, in vanilla gradient Simonyan et al. (2014) we have $S(g_c, x) = g_c(x)$ while integrated gradient Sundararajan et al. (2017) defines $S(g_c, x) = (x - x') \odot \int_0^1 g_c(x' + \alpha(x - x')) d\alpha$, where $x'$ is a predetermined baseline. Then, we substitute $g_c$ in the function $S$ with our likelihood ratio gradient estimator $\hat{g}_c$ defined in eq. (2), thereby enabling saliency map generation in the black-box setting, formalized as follows:

$$M_c^{LR}(f, x) = S(\hat{g}_c^{\text{LR}}, x). \tag{5}$$

### 3.3 BLOCKWISE COMPUTATION FOR SCALABILITY

**Challenges with estimation variance**. The aforementioned likelihood ratio method relies on the correlations between injected noise and final outputs to estimate the gradients of interested parameters

or the inputs. Nevertheless, as the model becomes more complex and the input dimension $d$ increases—a typical scenario in scaled-up models—the estimation variance, $\text{Var}([\hat{g}_c^{\text{LR}}]_i)$, $i \in \{1, ..., d\}$, becomes unbearable. This makes it difficult to implement in real-world applications, notably including the craft of saliency maps. Despite the requirement for only coarse gradients, high estimation variance in saliency maps associated with large images can yield vastly inconsistent attribution conclusions.

**Enhancing variance reduction through blockwise estimator** To make it scalable for interpreting the model decisions on high dimensional inputs, we further introduce a blockwise adaption of the likelihood ratio method for saliency map generation. Concretely, this approach involves initially selecting a small segment (referred to as a "block") randomly on the original image. We adopt the sampling that ensures each pixel in the image has an equal probability $q$ of being covered by the block. Then, we inject noise exclusively into the block area and leave the remainder of the image unchanged. Subsequently, we calculate the average of likelihood ratio proxy gradients across multiple random blocks and noise instances to form the resulting blockwise likelihood ratio gradient estimator, formalized as follows:

$$\hat{g}_c^{\text{BLR}}(x_0) := \frac{1}{nq} \sum_{i=1}^{n} (-f_c(x_0 + J_i \odot z_i) \nabla_z \ln \mu_z(J_i \odot z_i)), \tag{6}$$

where $\{J_i\}_1^n$ represents the masks for random blocks—specifically, $[J_i]_k = 1$ if the pixel $k$ is covered by the block and 0 otherwise. Similarly, when $\{z_i\}_1^n$ are Gaussian noise, we can simplify the blockwise likelihood ratio gradient estimator into $\hat{g}_c^{\text{BLR}}(x_0) = \frac{1}{nq\sigma^2} \sum_{i=1}^{n} (f_c(x_0 + J_i \odot z_i) J_i \odot z_i)$.

Under the same asymptotic growth condition, we derive the following theorem 2 for the blockwise likelihood ratio gradient estimator. This theorem leads us to a direct but critical corollary, $\lim_{\sigma \to 0^+} \mathbb{E}(\hat{g}_c^{\text{BLR}}) = g_c(x)$, ensuring that blockwise estimator's efficacy in accurately estimating the gradient of black-box models with controlled standard deviation of integrated noise. The proof of the corollary is provided in detail in appendix B.1.

**Theorem 2** *Assume that $\lim_{|\zeta| \to \infty} f_c(x_0 + \zeta) \mu_z(\zeta) = 0$ for any input $x_0$. Then, for any noise $z$ that is independent between dimensions, we have*

$$\mathbb{E}(\hat{g}_c^{BLR}(x_0)) = \mathbb{E}_{z,J}(\frac{1}{q} J \odot g_c(x_0 + J \odot z)). \tag{7}$$

**Proof 2** *Notice that when noise $z$ is independent between dimensions, we have $\nabla_\zeta \frac{\mu_z(\zeta)}{\mu_z(J \odot \zeta)} = 0$. Then, we can obtain*

$$\begin{aligned}
\mathbb{E}(\hat{g}_c^{BLR}(x_0)) &= \mathbb{E}_{z,J}(-\frac{1}{q} f_c(x_0 + J \odot z) \nabla_z \ln \mu_z(J \odot z)) \\
&= \mathbb{E}_J \left( -\frac{1}{q} \int_{\mathbb{R}^d} f_c(x_0 + J \odot \zeta)(\nabla_\zeta \mu_z(J \odot \zeta)) \frac{\mu_z(\zeta)}{\mu_z(J \odot \zeta)} d\zeta \right) \\
&= \mathbb{E}_J \left( -\frac{1}{q} \int_{\mathbb{R}^d} f_c(x_0 + J \odot \zeta) \nabla_\zeta \mu_z(\zeta) d\zeta \right) \\
&= \mathbb{E}_J \left( \frac{1}{q} \int_{\mathbb{R}^d} (J \odot \nabla_x f_c(x)|_{x_0 + J \odot \zeta}) \mu_z(\zeta) d\zeta \right) \\
&= \mathbb{E}_{z,J}(\frac{1}{q} J \odot g_c(x_0 + J \odot z)).
\end{aligned} \tag{8}$$

While both the standard and blockwise likelihood ratio estimators are equivalent in terms of expected value, the implementation of blockwise computation introduces a significant advantage in variance reduction across each dimension, if the number of times that noise is added to that dimension is maintained. Formally, given the same number of copies for each input dimension in the gradient estimator computed in the standard manner in eq. (2) and the blockwise manner in eq. (6), we have

$$\text{Var}([\hat{g}_c^{\text{BLR}}(x_0)]_i) < \text{Var}([\hat{g}_c^{\text{LR}}(x_0)]_i), \ \forall i \in \{1, ..., d\}. \tag{9}$$

The proof and detailed analysis can be found in appendix B.2. This property supports us in reducing the gradient estimation variance for saliency map generation. We define our blockwise likelihood ratio method for saliency map generation in the same way as eq. (5) but leveraging $\hat{g}_c^{\text{BLR}}$.

### 3.4 THE SELECTION OF THE CLASS ACTIVATION $f_c$

In the previous discussion, a gradient-based saliency map harnesses the gradient of the class activation w.r.t. inputs, $g_c(x_0) = \nabla_x f_c(x)|_{x=x_0}$, to explain the decision of neural networks. While we have addressed the "undifferentiable" problem due to inaccessibility for black-box models by utilizing the likelihood principle mentioned above, there remains an open question: *what if we can't access the class activation $f_c$ of black-box models?* Here, we propose solutions for three possible scenes in real-world applications.

**Soft- & Hard-label decision**. When logits or confidence of models are given, we can directly use them to estimate the gradient of the loss function, *i.e.,* cross-entropy, to the inputs for saliency map generation in our framework. When only the predicted label is given, *i.e.,* the hard label, we can not directly compute the loss function for gradient estimation. To address this issue, we propose to compute the nearest path distance between the predicted class and ground truth in WorldNet as the surrogate criteria to evaluate the classification performance.

**Text-based decision**. (Multimodal-) language models output the texts for users' queries, making it difficult to define the performance criteria for gradient estimation directly. To address this problem, we propose a prompt construction strategy. Specifically, when querying the decision of one image, we will additionally ask the model to respond to our answer by following the format of "The input image belongs to [MASK]", where the mask is filled by taking one category of WordNet. Then, we will compute the nearest distance between two nodes, *i.e.,* one that comes from the model and the ground truth, for performance evaluation and gradient estimation.

## 4 EXPERIMENT

In our experiments, we set up three different tasks to verify the use of forward gradient in explaining model decisions: the saliency map generation for computer vision models, sentiment analysis for language models, and bias interpretation for vision-language models. For a vision-involved explanation, we leverage $5,000$ fine-grained noisy blocks to estimate the forward gradient. For text explanation, we leverage $1,000$–$96,000$ noisy copies of the original inputs to estimate the forward gradients. We use the auto-grad function provided by the PyTorch library to compute the vanilla gradient as the white-box explanation. All experiments are conducted in a single A6000 GPU with 48 GB VRAM.

### 4.1 SALIENCY MAP GENERATION ON VISUAL MODELS

We first evaluate the performance on the task of generating the saliency maps on the ImageNet-1K dataset (Deng et al., 2009). We compare the forward gradient with the vanilla gradient. The vanilla gradient is fetched by the auto-grad in the PyTorch library, while the forward gradient has no access to the model information except the input and output.

**Visualization**. We present the results in fig. 3. It can be observed that all saliency maps generated by the vanilla gradient are more susceptible to inherent noise in the image, indicating weaker interpretability in explaining model decisions. In contrast, the forward gradients naturally smooth out the noise by injecting perturbations, thereby providing better explainability. This can be evidenced by the main objects are significantly highlighted on the saliency maps generated by the forward gradient compared with the vanilla gradient. Also, the saliency maps generated for different models by the forward gradients also depict large difference. While only one fish is highlighted in explaining the AlexNet, nearly all fishes are highlighted in explaining the ResNet-50. This demonstrates the good scalability of the forward gradient in explaining various architectures.

**Deletion&Insertion**. Beyond the visualization results, we also use the deletion and insertion game to demonstrate the effectiveness of the forward gradient in explaining the models. In the deletion task, we remove the most important features and observe the drop in the model's confidence, indicating the crucial role of these features—*where a lower deletion score is better*. In the insertion task, we reintroduce the important features into a blank image and track the increase in the model's confidence, with *a higher insertion score indicating better performance*.

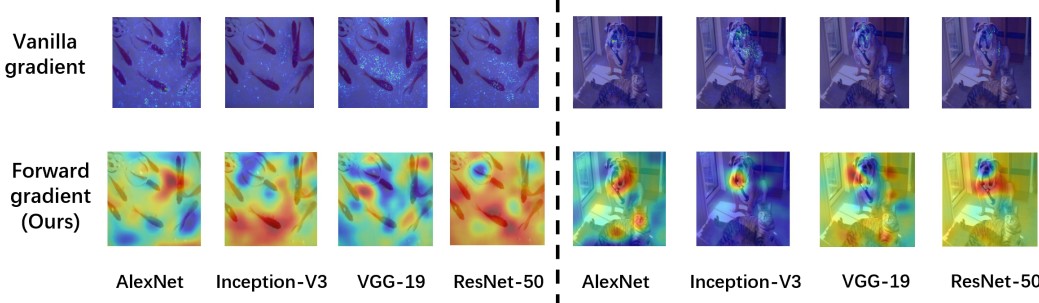

Figure 3: Visualization comparison between the saliency maps of four different models generated by vanilla gradient and forward gradient (ours). The lightness indicates the strength of model's interest.

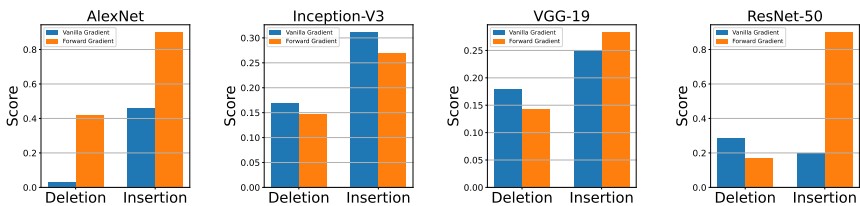

Figure 4: Quantitative comparison between the vanilla gradient and forward gradient in explaining model decisions. We play the deletion (↓)&insertion (↑) game on four models to compare the performance.

The quantitative comparison results are shown in fig. 4. In general, though without any access to the model architecture, the forward gradient outperforms the vanilla gradient in explaining three of four models (AlexNet×, Inception-V3✓, VGG-19✓, ResNet-50✓), as indicated by the scores. Dive into the results of AlexNet, we can find the cause of failure from fig. 3, where the highlighted area has some position shift to some degree with the main object, which could be due to the estimation variance in gradient estimation. On the other hand, Although the vanilla gradient achieves a better score for AlexNet, the highlighted areas in fig. 3 are not sufficiently explainable. This could be attributed to adversarial perturbations rather than the semantic areas indicating the model's interests.

## 4.2 EXPLAINING TEXT CLASSIFICATION MODELS ON SENTIMENT ANALYSIS

In this study, we delve deeper into the interpretability of the forward gradient method in natural language processing (NLP) tasks. Specifically, we train an attention-based bidirectional LSTM model for text classification using the Stanford Sentiment Treebank-2 (SST-2) dataset (Socher et al., 2013). The model incorporates pre-trained GloVe embeddings for word representation and achieves an overall accuracy of $82.4\%$ on the test set. We focus on explainability by analyzing gradients with respect to the GloVe word embeddings rather than the word indices. Consistent with previous experimental setups, we compare the forward gradient method's performance with the vanilla gradient, computed using PyTorch's Auto-grad feature, to highlight differences in interpretability and insights.

As shown in fig. 5, we present the analysis of four sentences. The word importance identified by the vanilla gradient and forward gradient shows high consistency, differing primarily in magnitude. For example, in the first sentence, both gradients indicate that the word 'must' plays a crucial role in the final decision. However, a notable difference is observed in their treatment of the word 'believed.' While the vanilla gradient assigns it a less significant role, the forward gradient analysis suggests its considerable importance in classifying the sentence as positive. We argue that this observation aligns more closely with human understanding of natural language processing tasks.

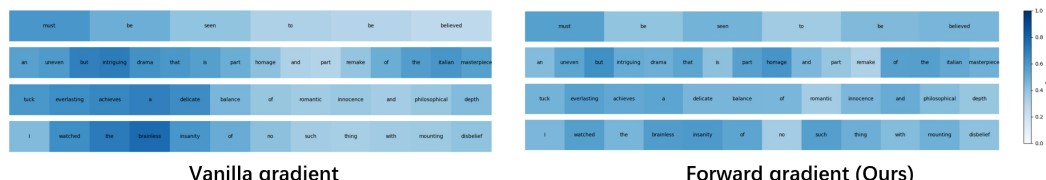

Figure 5: Comparison of vanilla gradient and forward gradient in explaining the text models from the word embedding level. A darker color indicates a higher importance approved by the gradient.

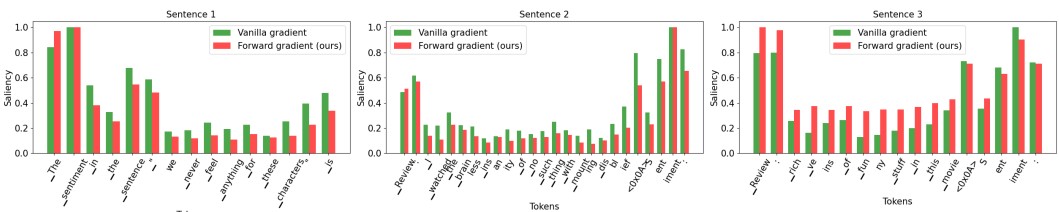

Figure 6: Comparison of calculated saliency scores and estimated saliency scores on LLM. Our method shows the same trend of variation as the calculated saliency scores.

### 4.3 EXPLAINING THE PHI-3

We evaluated our method on Phi-3 mini(Abdin et al., 2024). We prompt the LLM to determine the sentiment of three movie reviews obtained from the SST-2 dataset(Socher et al., 2013). The saliency map estimated by our method aligns with the overall trend observed in the saliency map generated using the vanilla gradient method. However, the noise introduced in our estimation process makes it less contrastive, meaning there is a less pronounced difference in saliency score between tokens with high and low saliency scores. Nevertheless, this effect can be mitigated by subtracting a bias from the saliency scores and rescaling the map. As shown in fig. 6, our estimated saliency map closely aligns with the saliency map generated using the vanilla gradient method.

### 4.4 BIAS INTERPRETATION FOR VISION-LANGUAGE MODELS

We move on to a more challenging task, using the forward gradient to explain vision-language models on a biased dataset. Specifically, we focus on the CLIP model, employing the ViT-B/32 architecture as the feature encoder. The dataset used for this study is Hard ImageNet, which contains many spurious correlations that can mislead the model's decision-making. In the absence of explicit feedback signals from downstream tasks, we use the cosine similarity between the target class label and the image features to estimate the forward gradient.

We select five classes, including the howler monkey, seat belt, space bar, dog sled, and balance beam, and present the interpretation results in fig. 7. We provide the text embeddings with the ground-truth labels and compute the forward gradient as previously described to identify which parts contribute to the model's decision. From the results, we observe that the CLIP encoder performs well without bias

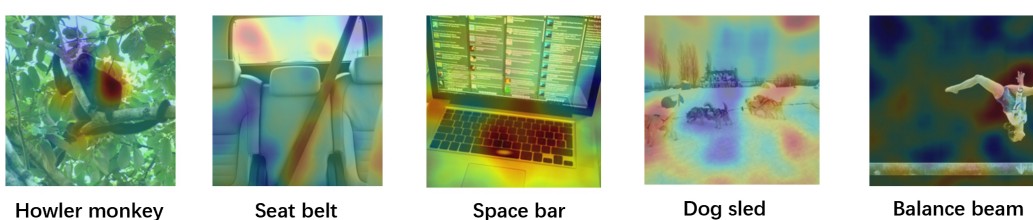

Figure 7: Leverage the forward gradient to explain the CLIP on Hard-ImageNet.

for the 'howler monkey' and 'space bar' classes. However, it is influenced by spurious correlations in the remaining three classes. For instance, while CLIP correctly identifies 'seat belt,' it also associates the concept with the outside sky. In the case of the 'dog sled' class, CLIP links the concept primarily with the dog, neglecting the sled. A similar phenomenon occurs with the 'balance beam' class, where the model focuses mainly on the athlete, causing the balance beam to be overlooked, resulting in misalignment between the visual and text embeddings. On the one hand, these results reveal that CLIP is susceptible to spurious correlations, even when trained on a large-scale, multi-modal dataset. On the other hand, we can see that the forward gradient serves as a powerful tool for analyzing and understanding failures in black-box models in practical applications.

### 4.5 ABLATION STUDY

we provide an ablation study on the involved gradient estimation part. On text models, we use the cosine similarity between the estimated gradient and vanilla gradient to study the impact of the number of word embedding copies on estimation accuracy and use the insertion score to study the influence of block-wise computation in explaining vision models.

Table 1: Gradient estimation accuracy by varying the number of copies.

| # copies | 10 | 20 | 30 | 40 | 50 | 100 |
|---|---|---|---|---|---|---|
| Similarity (%) | 13.1 | 19.7 | 23.3 | 24.2 | 29.6 | 36.3 |

Table 2: Insertion score using the blockwise technique with different numbers of blocks.

| # Blocks ($\times 10^3$) | 1 | 2 | 3 | 4 | 5 |
|---|---|---|---|---|---|
| Insertion score (%) | 51.33 | 59.54 | 60.08 | 60.26 | 57.20 |

**The gradient estimation accuracy using likelihood ratio method**. On a text model, we vary the number of copies for word embedding to study the performance. As shown in Tab. 1, we can see that the accuracy of the gradient estimation is improved when the number of word embedding copies is increased. While it only has a similarity of $13.1\%$ for the estimated gradient with the ground-truth, it can be improved to $36.3\%$ when taking 100 copies into the estimation process. Thus, there is a trade-off between computation requirement and robust interpretation of model decision.

**The blockwise computation for variance reduction**. We report the ablation study of the use of blockwise computation in Tab. 2. We vary the number of small blocks in gradient estimation from $1,000$ to $10,000$. The size of blocks is set to $10\%$ of the original image size. With increasing the number of blocks, the insertion score can be improved, indicating a better capacity in explaining models. However, with large enough number of blocks, the gradient estimation can get stuck in a local optima due to the balance between estimation redundancy and variance. Specifically, it achieves the peak performance when selecting $4,000$ as the number of blocks in computation, while a larger number copies $5,000$ causes a performance degradation of $3.06\%$.

## 5 CONCLUSION

The likelihood ratio method makes it possible to train neural networks with only forward passes. Inspired by this, we design a pipeline to employ the likelihood ratio method in gradient estimation, particularly for interpreting model decisions in black-box scenarios. To address the issue of large gradient estimation variance, we propose a blockwise computation technique that balances computational cost with task performance. We explore the potential of the forward gradient in explaining models across three tasks: saliency map generation, sentiment analysis on text models, and interpreting spurious correlations learned by vision-language models. The experimental results demonstrate the effectiveness and scalability of the forward gradient in explaining black-box models. This study paves the way for future research in explaining black-box models.

### LIMITATIONS

While the forward gradient offers a method to estimate gradients without relying on the chain rule—facilitating the explanation of black-box models based on gradient information—it is computationally expensive. For larger models, this approach may require thousands of iterations to achieve an acceptable estimation. Developing effective variance reduction methods is crucial not only for training models without backpropagation (BP) but also for improving the explainability of black-box models.

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

## A    ALGORITHM

We present the algorithm details in 1. To explain the decision of the black-box model $f$ on the inputs $x$, we first generate $n$ copies as $\hat{x}$. Then, we sample $n$ independent random noise from the normal distribution to perturb the copies. Next, with the target label $y$, we compute the loss function value for each noisy copy and get the proxy gradient. Last, we compute the average of proxy gradients, get the estimated gradient, and employ the white-box saliency map generation method to craft $s$.

---

**Algorithm 1:** Likelihood ratio method for saliency map generation

---

**input**   : Image $x$ with label $y$, black model $f$, loss function $\mathcal{L}(\cdot, \cdot)$, number of copies $n$,
          while-box saliency map generation method $S(\cdot)$
**output** : The saliency map $s$ to interpret the decision of $f$ on $x$

1  $\hat{x} \leftarrow$ *generate $n$ copies of $x$*;
2  $\delta \leftarrow$ *sample $n$ i.i.d. noise from the normal distribution*;
3  $\hat{x} \leftarrow \hat{x} + \delta$;
   // perturb the inputs
4  $\hat{l} \leftarrow \mathcal{L}(f(\hat{x}), y)$;
5  $\hat{g} \leftarrow \hat{l} \cdot \delta$;
   // compute the proxy gradient
6  $g \leftarrow \frac{1}{n}\hat{l}_{i=1}^{n}$;
   // compute the estimated gradient
7  $s \leftarrow S(g)$

---

## B    THEORY ANALYSIS

### B.1    PROOF OF THE COROLLARY

We provide the proof of the corollary we present in 3.3. Mathematically, the corollary is formalized as

**Corollary 1** *Assume the model's gradient $g_c$ is bounded and let the clockwise likelihood ratio gradient estimator $\hat{g}_c^{BLR}$ as defined in 6. Then for any input $x_0$, we have*

$$\lim_{\sigma \to 0^+} \mathbb{E}(\hat{g}_c^{BLR}(x_0)) = g_c(x_0) \tag{10}$$

**Proof 3** *Consider arbitrary Gaussian noise r.v. sequence $\{z_i\}_1^\infty$ that satisfies $\mathbb{E}(z_i) = 0$ and $\lim_{i \to \infty} \Sigma_i = 0$.*

*It follows that*

$$z_i \xrightarrow{L_2} 0$$

$$\Rightarrow z_i \xrightarrow{P} 0$$

$$\Rightarrow J \odot z_i \xrightarrow{P} 0$$

$$\Rightarrow x_0 + J \odot z_i \xrightarrow{P} x_0$$

$$\Rightarrow g_c(x_0 + J \odot z_i) \xrightarrow{P} g_c(x_0)$$

$$\Rightarrow \frac{1}{q} J \odot g_c(x_0 + J \odot z_i) \xrightarrow{P} \frac{1}{q} J \odot g_c(x_0) \tag{11}$$

$$\Rightarrow \frac{1}{q} J \odot g_c(x_0 + J \odot z_i) \xrightarrow{L_1} \frac{1}{q} J \odot g_c(x_0)$$

$$\Rightarrow \lim_{i\to\infty} \mathbb{E}_{z_i,J}(\frac{1}{q} J \odot g_c(x_0 + J \odot z_i)) = \mathbb{E}_{z_i,J}(\frac{1}{q} J \odot g_c(x_0))$$

$$\Rightarrow \lim_{i\to\infty} \mathbb{E}_{z_i,J}(\frac{1}{q} J \odot g_c(x_0 + J \odot z_i)) = g_c(x_0)$$

*Recall that by 2, we have $\mathbb{E}(\hat{g}_c^{BLR}) = \mathbb{E}_{z,J}(\frac{1}{q} J \odot g_c(x_0 + J \odot z))$. This, together with the arbitrariness of $\{z_i\}_1^\infty$ as long as $\lim_{i\to\infty} \Sigma_i = 0$, indicates that*

$$\lim_{\sigma\to 0^+} \mathbb{E}(\hat{g}_c^{BLR}(x_0)) = g_c(x) \tag{12}$$

## B.2 ANALYSIS OF VARIANCE REDUCTION FROM THE BLOCKWISE COMPUTATION

In this discussion, we consider the Gaussian noise $z \sim N(0, \sigma^2\mathbb{I})$. Other situations are similar. Then, we have simplified estimators as follows:

$$\hat{g}_c^{\text{LR}}(x_0) = \frac{1}{n^{\text{LR}}\sigma^2} \sum_i^{n^{\text{LR}}} f_c(x_0 + z_i)z_i \tag{13}$$

$$\hat{g}_c^{\text{BLR}}(x_0) = \frac{1}{n^{\text{BLR}}q\sigma^2} \sum_{i=1}^{n^{\text{BLR}}} (f_c(x_0 + J_i \odot z_i)J_i \odot z_i). \tag{14}$$

Given the same number of copies for each input dimension in the gradient estimator computed in the standard manner, *i.e.*, $n^{\text{LR}} = n^{\text{BLR}}q$, we need to prove $\text{Var}([\hat{g}_c^{\text{LR}}(x_0)]_i) < \text{Var}([\hat{g}_c^{\text{BLR}}(x_0)]_i)$, $\forall i \in \{1, ..., d\}$. To prove this, we only need to prove $\text{Var}([f_c(x_0 + z)z]_i) < \text{Var}([f_c(x_0 + J \odot z)J \odot z]_i \mid [J]_i = 1)$. Using multivariate Taylor expansion, when $\sigma$ is small, we can say

$$f_c(x_0 + z) = f_c(x_0) + \nabla_x f|_{x=x_0} \cdot z. \tag{15}$$

It follows that

$$\mathbb{E}(f_c(x_0 + z)z) = 0 + \nabla_x f|_{x=x_0}\sigma^2\mathbb{I} = \sigma^2 \nabla_x f|_{x=x_0}. \tag{16}$$

Then, we have

$$\text{Var}([f_c(x_0 + z)z]_i) \tag{17}$$

$$= \mathbb{E}([f_c(x_0 + z)z]_i^2) - \sigma^4[\nabla_x f|_{x=x_0}]_i^2 \tag{18}$$

$$= \mathbb{E}(f_c^2(x_0 + z)[z]_i^2) - \sigma^4[\nabla_x f|_{x=x_0}]_i^2 \tag{19}$$

$$= \mathbb{E}((f_c(x_0) + \nabla_x f|_{x=x_0} \cdot z)^2[z]_i^2) - \sigma^4[\nabla_x f|_{x=x_0}]_i^2 \tag{20}$$

$$= f_c^2(x_0)\sigma^2 + 2\sigma^4[\nabla_x f|_{x=x_0}]_i^2 + \sigma^4 \sum_{k=1, k\neq i}^{d} [\nabla_x f|_{x=x_0}]_k^2 \tag{21}$$

Similarly, we have

$$\mathbb{E}(f_c(x_0 + J \odot z)J \odot z \mid [J]_i = 1) = \sigma^2[\nabla_x f|_{x=x_0}]_i \tag{22}$$

and then

$$\text{Var}([f_c(x_0 + J \odot z)J \odot z]_i \mid [J]_i = 1) \tag{23}$$

$$= \mathbb{E}([f_c(x_0 + J \odot z)J \odot z]_i^2 \mid [J]_i = 1) - \sigma^4[\nabla_x f|_{x=x_0}]_i^2 \tag{24}$$

$$\approx f_c^2(x_0)\sigma^2 + 2\sigma^4[\nabla_x f|_{x=x_0}]_i^2 + q\sigma^4 \sum_{k=1, k \neq i}^{d} [\nabla_x f|_{x=x_0}]_k^2 \tag{25}$$

$$< \text{Var}([f_c(x_0 + z)z]_i), \tag{26}$$

if $\exists k \in \{1, ..., d\}$, $k \neq i$, such that $[\nabla_x f|_{x=x_0}]_k \neq 0$.

