# OpenReview forum: "Is Forward Gradient an Effective Tool for Explaining Black-box Models?"
_ICLR.cc/2025/Conference — Submitted to ICLR 2025_

### Official Review · Reviewer_GBTB · 2024-10-28

**Soundness:** 2
**Presentation:** 1
**Contribution:** 1
**Rating:** 3
**Confidence:** 5

**Summary:**

This paper borrows the likelihood ratio method from the gradient estimation literature and introduces a method to explain decisions by black boxes. The motivation for adopting estimated gradients as an alternative to actual gradients is somewhat clear — it can be difficult to measure gradients exactly in certain scenarios. Furthermore, the authors acknowledge the challenge of estimation variance and propose a countermeasure. The proposed method is tested across various settings and evaluated primarily through qualitative analysis. Two major concerns relate to the novelty of the method and the clarity of some technical details.

**Strengths:**

1. The authors spent efforts on analysis to provide theoretical support for the proposed method.
2. The authors targeted the well-observed issue of estimation variance and tried to propose a countermeasure.

**Weaknesses:**

1. The novelty of the gradient estimation method warrants further clarification. While the authors state that they "propose" the likelihood ratio gradient estimator, there is no derivation or clarification about how the exact form of Equation 2 is reached. In fact, the formula closely resembles the well-known likelihood ratio (also called score-function) gradient estimator, with only minor differences in notation from Equation 13d in the survey paper [1]. Furthermore, the idea of adopting gradient estimation for feature attribution has been explored by [2].
2. The analysis of the variance reduction effect achieved by the blockwise estimator seems to be questionable. In Appendix.B.2, the authors complete the proof of variance reduction under the condition that $n^{\textrm{LR}}=n^{\textrm{BLR}}q$. However, given $0< q < 1$, the condition implies that the BLR estimator uses a larger number of copies than the LR estimator, thereby putting the increased number of forward passes as the more likely source of the reduced variance.
3. Major improvements could be made in the writing and overall organization of the paper, particularly in the introduction, related works, and experiment sections.
4.  The observations on the experimental results rely heavily on qualitative analysis, which appears quite subjective and less convincing.

[1] Mohamed, Shakir, et al. "Monte Carlo Gradient Estimation in Machine Learning." *Journal of Machine Learning Research* 21.132 (2020): 1-62. \
[2] Cai, Yi, and Gerhard Wunder. "On Gradient-like Explanation under a Black-box Setting: When Black-box Explanations Become as Good as White-box." *In International Conference on Machine Learning, pp. 5360–5382. PMLR, 2024.*

**Questions:**

Regarding the methodology:
1. Could the authors clarify how the gradient estimator described by Equation 2 differs from previous work?
2. The claim in line 91 about gradient estimation having "greater scalability" than true gradients is a bit unclear. Given that estimation variance increases significantly when the input feature space expands, should the claim instead be the opposite? Exact gradient measurement (e.g. through backpropagation) is barely affected by the dimensionality of the feature space.
3. The statements regarding the unbiasedness of the LR estimator seem inconsistent. Lines 65-66 state "allowing for __unbiased__ gradient estimation", whereas lines 251-252 say that the LR estimator "introduces a certain __bias__".
4. Could the authors elaborate more about the condition in Proof 2? Specifically, why does $\nabla_\zeta \frac{\mu_z(\zeta)}{\mu_z(J\odot\zeta)}=0$ hold? How does this condition facilitate the derivation in Equation 8, particularly the equality between the second and the third lines?

Regarding the writing and experiments:

1. As a main concept defining the problem to be solved, how is the black-box setting defined? What is the relationship between "black-box models" and "black-box settings"?
2. Some statements appear contradictory and confusing, for example:
    - Lines 51-53 state that gradient-based methods can explain "decisions from black-box models", whereas line 161 claims that "… the reliance on these gradients poses a significant challenge in the context of black-box models, …".
    - How many forward passes are needed to derive an explanation? There is the claim in Figure 1 "Our proposed method only needs one forward pass of the studied model", yet line 353 states "…, we leverage 1,000 - 96,000 noisy copies …".
3. How are "the most important features" defined in the deletion & insertion test? What is the deletion score, and how is it computed?
4. What is the target class for the examples shown in Figure 7? Does the coloring have a specific meaning?
5. The visual examples in Figures 7 and 10 are somewhat difficult to comprehend. Without a comparison to explanations by other competitors, it is difficult to determine whether the issue lies with the CLIP model or with the explainer itself.

---

### Official Review · Reviewer_xGD3 · 2024-11-03

**Soundness:** 2
**Presentation:** 3
**Contribution:** 2
**Rating:** 3
**Confidence:** 4

**Summary:**

The paper proposes to use the forward-gradient method to estimate the gradient of black-box predictive models and use these as plug-in estimators in gradient-based saliency maps for explaining the model decisions. It introduces the likelihood ratio gradient estimator based on perturbing the original data and forward passing these through the NN model showing that with decreasing variance of the noise this converges in expectation to the true gradient. It further proposes to reduce the variance of the estimator through a block-wise masking scheme for the noise injection. The method is evaluated through presenting the explanatory saliency maps and results of drop/insert game over models for image classification, text sentiment analysis, and vision-language models claiming superior explanatory value over standard back-prop gradient counterparts. The limitation of the method in terms of computational costs due to the need of multiple noise resamples is duly noted.

**Strengths:**

The authors propose to combine the recent forward-gradient estimator approach with gradient-based approaches for explaining model decisions for black-box models and thus solve the problem of unavailability of standard back-prop gradient. Their theoretical analysis shows the convergence of this estimator to the true gradients and thus justifies its use as a plug-in replacement in the gradient-based explainability methods. The proposed block-wise noise injection scheme is shown to reduce the variance and thus speed up the convergence of the estimator reducing the computational costs. The paper is well written and easy to follow.

**Weaknesses:**

- The paper focuses on the gradient-based approach to explainability and strives to extend this to the black-box situation where access to inner workings of the model is not available. The proposed method relies on data perturbations to estimate the gradient. Various types of data perturbation methods have previously been proposed in the XAI literature for black-box model set-up (see e.g. Arrieta, Alejandro Barredo, et al. "Explainable Artificial Intelligence (XAI): Concepts, taxonomies, opportunities and challenges toward responsible AI." Information fusion 58 (2020): 82-115.), some even discussing Gaussian perturbations (e.g. Fong, Ruth C., and Andrea Vedaldi. "Interpretable explanations of black boxes by meaningful perturbation." Proceedings of the IEEE international conference on computer vision. 2017.). The current paper completely disregards these when discussing related work and hence it is not clear to what degree it is innovative and/or similar to previous work in this area.
-  Adding small amounts of Gaussian noise to change the label prediction is similar to adversarial attacks. It has been previously recognized (Ghorbani, Amirata, Abubakar Abid, and James Zou. "Interpretation of neural networks is fragile." Proceedings of the AAAI conference on artificial intelligence. Vol. 33. No. 01. 2019.; Kindermans, Pieter-Jan, et al. "The (un) reliability of saliency methods." Explainable AI: Interpreting, explaining and visualizing deep learning (2019): 267-280.) that as long as the classifier has not been trained in a robust way w.r.t. such perturbations, its gradient information is unreliable for gradient-based explanations. In consequence it may be assumed that the perturbation method proposed in the current paper may suffer from similar problems. This issue is, however, not discussed at all in the paper.
- The conclusions from the experimental evaluation seems counterintuitive or even contradictory to the theoretical analysis presented in the paper. The theoretical analysis endeavors to come up with an forward-based gradient estimator that would be "good-enough" estimator of the true back-prop gradient. It proves its convergence in expectation and further strive to reduce its variance to speed up its convergence to the true back-prop gradient. Yet, the experiments seem to claim that using these estimated gradients actually has even better explanatory value than the back-prop gradient. It is not clear to me, how this could be possible given the theoretical claims. This for me is a significant logical failure that needs to be either better explained or rectified.
- The experimental evaluation is not convincing due lack of provided detail.
    - The saliency maps in Figure 3 are difficult to interpret due to missing scale (missing color-map). Are these the absolute values of the gradients or signed gradients? Are these rescaled the same across the images?
    - Simple back-prop gradients are typically not considered to have high explanatory value in the XAI literature (see lit above). Instead, they are a classical method of obtaining adversarial examples. It is therefore not clear, what is the take-home message from Figure 3.
    - It is not clear how exactly is the drop/insert game executed. Is the same number of features dropped / inserted in all cases or is it rather based on some threshold to decide the "most important" features. Also the conclusions are not clear - the text claims that forward gradient outperforms simple gradient on Inception-V3 while the first has lower insert score than the letter. This does not seem correct.

**Questions:**

1. See the Weaknesses section.
2. In the three key findings in the Intro section the paper claims "comparable interpretability with greater scalability". What do you mean by greater scalability with the view of the acknowledged limitation of high computation cost of the method?
3. Why is the method called likelihood-ratio? Where / what is the link to likelihood-ratio?
4. What are the input-output spaces of M and S in equation (1). Please complement the notation.
5. In the beginning of section 3.2 the framework is indicated as "unified". In what sense is it unified?
6. The theoretical proves are derived for $\sigma \to 0$. What is the noise variance used in practice in the experiments? Have you explored different noise levels? What was the effect? It would be good to see this analysis in an ablation study.
7. In section 4.1 in paragraph Visualization you claim that the method highlights only one fish for AlexNet and nearly all fish for ResNet and this is a sign of scalability of the method. What do you mean? What does the number of fish has to do with the method scalability?
8. In the last sentence in page 8 you "argue" that your observation aligns well with human understanding. How can you back this argument? Has it been previously shown anywhere or have you verified it yourself (e.g. through human study)?
9. In section 4.4. you present examples over five selected classes. How have these been selected? Cherry-picked?

---

### Official Review · Reviewer_wF1E · 2024-11-03

**Soundness:** 2
**Presentation:** 3
**Contribution:** 1
**Rating:** 3
**Confidence:** 5

**Summary:**

The paper investigates the eligibility of forward-gradients computation for explaining blackbox models. The topic is quite important in my view since most practical settings are blackbox. Also the paper is well written although of significant theoretical depth. Having said this and maybe I'm missing a point here most of this work has been done already in the context of evolution strategies. Please check out this ICML 24 paper here:
https://arxiv.org/pdf/2308.09381
In this paper eqn. 1 matches with eqn. 2 in the present paper. The authors have done a great deal to analyze this setting way beyond the present paper, including the estimator. Particularly, the actual explanation methodology is much more general.

Altogether I think th authors should first check that line of work first.

**Strengths:**

The paper is well written and the topic is interesting.

**Weaknesses:**

There is no paper novelty. The literature review is insufficient and needs to be adapted.

**Questions:**

I have no significant question. The outcomes are reasonable.

---

### Official Review · Reviewer_TGeu · 2024-11-04

**Soundness:** 2
**Presentation:** 2
**Contribution:** 2
**Rating:** 5
**Confidence:** 4

**Summary:**

This paper uses a likelihood ratio method (LR) to approximate the gradient of a model's output with respect to its input under a black box setting. In doing so, it enables gradient-based explanations without back-propagation. Comparisons to original back-propagation methods and some exposition on interpreting a LLN using the method are provided.

**Strengths:**

Interesting application of gradient approximation using likelihood ratio.
Generally understandable exhibition of method and block method of variance control.

**Weaknesses:**

Some comparisons to other methods and exhibition of the method's advantage are missing. The paper could be improved by further comparisons to proximally comparable methods and a more detailed exposition of the method's advantages, if they exist. See below for further details.

**Questions:**

It appears that the greatest advantage to the LR method is that it is able to provide gradients, and thus gradient based explanations, without using back-propagation, but only model querying. Major questions affecting the efficacy of the method:

- How does LR method compare in computational time and accuracy other black-box methods. A survey should be done, and comparisons should be made to classic numerical differentiation and any other promising competitors (cost, accuracy, etc.)
- This method is approximating the gradient. What is the computational cost to recover the true gradients/gradient methods with high accuracy over a variety of models?
- Where is the likelihood ratio in this method? The paper could be improved by giving a quick intro of a likelihood ratio and how the method employs the concept.

Regarding the improvements of the method:
To list other strengths of the method that I noted: 1) provides interpretability, 2) does not require any knowledge of internals of black box, 3) provides improved scalability, 4) explanations smoothed-out noise.

1) Insofar as you are just approximating interpretability methods in a black-box scenario, this is just fine. However, the experiments make it look like the LR method makes a rough and smoothed approximation of the gradients, that the gradients were not recovered with much accuracy, but then the paper claims that this is an improvement. Do you want to approximation the saliency maps, or make an improvement? Either the original gradients and saliency maps should be recovered with sufficient accuracy, or significantly more justification needs to be given on how a deviation from them is an improvement. Other popular methods (IG, GradCAM) spend significant effort to justify why their methods are an effective interpretability method; if they are not recovered but altered so as to look/perform strikingly different, more work should be put into justifying the change (experimentally and theoretically).
2) Fine and good, the major use case of the method. The paper could be strengthened by a small expositing the practical use of this in the beginning.
3) Not sure what scalability means here. Could benefit from being more precise, having more treatment and specificity, an explanation on why this is beneficial.
4) Are you saying that deterministic methods have noise, while random approximations have less? The paper could improve by being precise about what the noise is, how it is removed. Also, I see in some visualizations that some areas with lots of bright spots for the normal method are not lit up for the LR method. This may be a problem for the claim that LR "smooths out" the original method and needs to be discussed.

Some comments on improving experiments:
- Can you provide theoretical or experimental metrics on computational costs/accuracy of your method vs backpropagation and numerical integration? It would be advantageous if the experiments were varied enough to get a generalized understanding of the differences.

- Can you provide time/accuracy graphs showing the effects of noise SD, number of samples used in the estimation, and also explain further/try the "large number of copies" mentioned?

- You claim that the saliency tests are improved because they are less affected by noise and cover more of the feature. Can the author provide experimental results on how many samples/how small the noise has to be so that the actual gradients and saliency maps are recovered with small error? Runtimes and computing specs should be given.

- Method of gradient estimation (theorem 1) follows other works. Those works should be sited, and any differences or novel additions should be pointed out. Likewise for block method of variance reduction.

Minor Comments:
115 - integrate gradients
195 Fig. fig
224 missing a1/2?
Ref for corollary (which theorem is being applied) would be nice)
37 large difference(s)
Figure 3: which saliency maps are these? What number of samples are being used for the gradient estimation?

---

### Meta-Review · Area_Chair_UMAf · 2024-12-18

**Metareview:**

This paper proposes using forward gradient methods to explain black-box model decisions without requiring access to model internals or backpropagation. The paper uses likelihood ratios to estimate model gradients and techniques to control the variance of these estimators. The method is more interpretable compared to other approaches. The authors provide experiments cross multiple domains such as images, text, and VLMs.

The paper addresses an important problem, provides some theoretical analysis, addresses potential downsides (such as estimator variance) and is applicable across data modalities. Reviewers agreed the paper was well written.

Reviewers had concerns about the novelty of the work. The likelihood ratio estimator proposed has appeared in prior work and its application to interpretability has been studied before. Experiments lack of clear comparison to other black-box explanation methods and experimental details are sparse with a heavy reliance on qualitative results.

Following consensus of the reviews, I will recommend rejection.

**Additional Comments On Reviewer Discussion:**

Reviewers had initially negative reviews bringing up weaknesses written in the above section. Authors did not engage in the rebuttal.

---

### Decision · Program_Chairs · 2025-01-22

Reject